# Preliminary Evidence for a Relationship between Elevated Plasma TNFα and Smaller Subcortical White Matter Volume in HCV Infection Irrespective of HIV or AUD Comorbidity

**DOI:** 10.3390/ijms22094953

**Published:** 2021-05-07

**Authors:** Natalie M. Zahr, Kilian M. Pohl, Allison J. Kwong, Edith V. Sullivan, Adolf Pfefferbaum

**Affiliations:** 1Department of Psychiatry and Behavioral Sciences, Stanford University School of Medicine, Stanford, CA 94305, USA; kilian.pohl@stanford.edu (K.M.P.); adolf.pfefferbaum@sri.com (A.P.); 2Neuroscience Program, SRI International, Menlo Park, CA 94025, USA; edie@stanford.edu; 3Gastroenterology and Hepatology Medicine, Stanford University School of Medicine, Stanford, CA 94350, USA; ajk@stanford.edu

**Keywords:** alcoholism, comorbidity, soluble proteins, cytokines, chemokines, IL1β, TNFα, IP10, complete blood count, metabolic panel, liver, infection, brain, structural MRI, magnetic resonance imaging, neuroimmune, neuroinflammation

## Abstract

Classical inflammation in response to bacterial, parasitic, or viral infections such as HIV includes local recruitment of neutrophils and macrophages and the production of proinflammatory cytokines and chemokines. Proposed biomarkers of organ integrity in Alcohol Use Disorders (AUD) include elevations in peripheral plasma levels of proinflammatory proteins. In testing this proposal, previous work included a group of human immunodeficiency virus (HIV)-infected individuals as positive controls and identified elevations in the soluble proteins TNFα and IP10; these cytokines were only elevated in AUD individuals seropositive for hepatitis C infection (HCV). The current observational, cross-sectional study evaluated whether higher levels of these proinflammatory cytokines would be associated with compromised brain integrity. Soluble protein levels were quantified in 86 healthy controls, 132 individuals with AUD, 54 individuals seropositive for HIV, and 49 individuals with AUD and HIV. Among the patient groups, HCV was present in 24 of the individuals with AUD, 13 individuals with HIV, and 20 of the individuals in the comorbid AUD and HIV group. Soluble protein levels were correlated to regional brain volumes as quantified with structural magnetic resonance imaging (MRI). In addition to higher levels of TNFα and IP10 in the 2 HIV groups and the HCV-seropositive AUD group, this study identified lower levels of IL1β in the 3 patient groups relative to the control group. Only TNFα, however, showed a relationship with brain integrity: in HCV or HIV infection, higher peripheral levels of TNFα correlated with smaller subcortical white matter volume. These preliminary results highlight the privileged status of TNFα on brain integrity in the context of infection.

## 1. Introduction

In seeking neural substrates of cognitive impairments observed in Alcohol Use Disorders (AUD) [1], medical conditions such as alcohol-related liver damage [2] or nutritional deficiencies (e.g., for thiamine which can result in Wernicke Korsakoff Syndrome) [3] must be considered [4] as they are independently related to cognitive dysfunction [5,6]. In both liver damage and thiamine deficiency, proposed biomarkers of disease include proinflammatory cytokines and chemokines. There is substantial support for the presence of peripheral inflammation—including proinflammatory protein production—in the transition from healthy liver to steatosis, hepatitis, and cirrhosis in alcohol-associated liver disease [7,8,9,10,11,12,13,14,15,16]. There is also experimental support for elevated peripheral levels of proinflammatory mediators in animal models of thiamine deficiency [17,18,19] and in non-alcohol-associated Wernicke’s encephalopathy [20], including tumor necrosis factor α (TNFα), interleukin (IL)1β, and IL6 [20,21,22].

Individuals with AUD have also been found to have increased representation of immune- and inflammation- related genes in the brain [23,24,25] and elevated protein levels of the chemokine monocyte chemoattractant protein-1 (MCP-1) in several brain regions [26]. In vivo, withdrawal from alcohol is associated with higher cerebrospinal fluid (CSF) levels of MCP-1 [27]. Peripheral (plasma/serum) elevations in MCP-1, TNFα, IL6, and interferon γ-induced protein 10 (IP10) [28,29,30] have been associated with AUD severity [31] or alcohol craving at early abstinence [32].

Our previous work reported higher than control IP10 and TNFα plasma levels in abstinent AUD individuals seropositive for hepatitis C virus (HCV) infection and in individuals infected with the human immunodeficiency virus (HIV) [33]. Specificity of IP10 and TNFα elevations to HCV infection status in AUD and HIV was provided by correlations between IP10 and TNFα levels, HCV viral load, and indices of liver integrity including the albumin/globulin ratio (AGR), fibrosis score (FIB4), and aspartate aminotransferase/platelet count ratio (APRI) [33].

The current study was conducted to confirm these findings in a larger sample of HCV seropositive individuals and to extend previous results by testing the hypothesis that elevated plasma IP10 and TNFα levels would be associated with compromised brain volume. Relatively few neuroimaging studies have evaluated the effects of HCV per se on the brain. Structural Magnetic Resonance Imaging (MRI) studies in HCV (absent hepatic encephalopathy or cirrhosis) relative to healthy controls report deficits in frontal and occipital cortical thickness [34], insular and thalamic volumes [35], or no effects of HCV on regional gray matter volumes [36]. Individuals with HCV exhibit altered metabolism in basal ganglia and white matter regions on MR Spectroscopy [37,38,39,40,41], while Diffusion Tensor Imaging (DTI) has shown increased diffusivity in white matter tracts including inferior longitudinal fasciculus and inferior fronto-occipital fasciculus [42]. In the context of HIV infection, HCV comorbidity further compromises white matter (e.g., greater volume deficits [43]; increased diffusivity [44,45], but see [46]) or shows greater anterior cingulate volume deficits [47] relative to HIV alone; in AUD, HCV comorbidity shows greater effects on frontal volume loss relative to AUD alone [48,49]. The few relevant neuroimaging studies in psychiatric disease have shown associations between elevated peripheral cytokines and smaller regional brain volumes: TNFα with smaller frontal gray volume in bipolar disorder/major depressive disorder [50] and with smaller hippocampus in major depressive disorder [51]; and elevated plasma IP10 with smaller volumes of basal ganglia structures [52]. 

Given the extant literature, the current study tested the hypotheses that elevated plasma TNFα would be associated smaller frontal gray and white matter volume, and elevated plasma IP10 would be associated with smaller caudate, putamen, and pallidum volumes.

## 2. Results

### 2.1. Study-Participant Demographics

Table 1 presents demographic data for each of the 4 groups. The control and AUD groups were younger than the HIV and AUD+HIVgroups (*p* = 0.007). The 3 patient groups relative to the control group were less educated, had lower socio-economic status (SES) [53] and global functioning (i.e., GAF) [54], scored lower on the Wechsler Test of Adult Reading (WTAR) [55], and endorsed more depressive symptoms (as determined by the BDI-II) (all *p* ≤ 0.0001). The Veterans Aging Cohort Study (VACS) index, which predicts all-cause mortality, cause-specific mortality, and other outcomes in those living with HIV infection [56], was higher in the 2 HIV groups (HIV and AUD+HIV) than the control or AUD groups.

### 2.2. Demographic Differences in Soluble Protein Levels

Initial analysis considered the effects of age, sex, race, and BMI on soluble protein levels across the 4 diagnostic groups. Soluble proteins showing age effects across all participants irrespective of diagnosis (N = 321) included IFNγ (r = −0.15, *p* = 0.006), IP10 (r = 0.23, *p* < 0.0001), MCP-1 (r = 0.17, *p* = 0.002), RANTES (r = 0.22, *p* < 0.0001), and TNFα (r = 0.16, *p* = 0.004). Only MDC showed sex effects (lower in men, t = −3.8, *p* = 0.0002). Race was differentially associated with the levels of GRO (F_321_ = 7.0, *p* = 0.001) and MDC (F_321_ = 7.9, *p* = 0.0005): higher in black people than white or other races; IP10 (F_321_ = 8.5, *p* = 0.0003) and MCP-1 (F_321_ = 9.8, *p* < 0.0001): higher in black people than white; IFNγ (F_321_ = 9.2, *p* = 0.0001) and IL12P70 (F_321_ = 5.1, *p* = 0.006): lower in black people than white or other races; and IL12P40 (F_321_ = 5.2, *p* = 0.006): lower in black people than white. None of the soluble proteins showed relationships with BMI. Soluble proteins showing age, sex, or race effects were statistically adjusted to produce Z-scores with the mean value of each soluble protein added back in (i.e., RANTES and TNFα for age; GRO, IL12P40, and IL12P70 for race; MDC for sex and race; IFNγ, IP10, and MCP-1 for age and race).

### 2.3. Patient Group Differences in Soluble Protein Levels

Results of separate 4-group ANOVAs for each soluble protein are presented in Table 2. Only 3 analytes showed significant 4-group effects surviving Bonferroni correction: IL1β was lower in all 3 patient groups relative to the control group (F_320_ = 7.04, *p* = 0.0001); IP10 (F_320_ = 14.76, *p* < 0.0001) and TNFα (F_320_ = 14.23, *p* < 0.0001) were higher in the 2 HIV groups relative to the control group (Figure 1). Evaluation of these 3 soluble proteins as a function of comorbidity with HCV in the AUD group demonstrated there was no interaction between IL1β and HCV status, whereas IP10 and TNFα levels were high only in AUD individuals seropositive for HCV (Figure 2). Among those with HIV, IP10 and TNFα levels were higher than controls regardless of HCV status. Two-group differences (*t*-tests) in soluble protein levels by initial diagnoses (i.e., AUD, HIV, or AUD+HIV) are presented in Appendix A and by HCV status collapsed across the 3 patient groups in Appendix A.

### 2.4. Soluble-Protein Correlations

Evaluation of IL1β correlations included the total AUD group and the 2 HIV groups (*n* = 235); IP10 and TNFα correlations included only HCV positive individuals in the AUD group and the 2 HIV groups (*n* = 127). Levels of IL1β, IP10, and TNFα were first evaluated for their relations with the volumes of 5 brain regions (frontal cortex, subcortical white matter, caudate, putamen, and pallidum). The only soluble protein showing a relation with brain volumes was TNFα: smaller subcortical white matter volumes were related to higher levels of TNFα (*ρ* = −0.24, *p* = 0.0095, Figure 3a). This relationship was also evident in HCV only (i.e., collapsed across the 3 initial groups, *n* = 57) subgroup (*ρ* = −0.36, *p* = 0.006, Figure 3b).

Further analysis considered relationships between ILβ, IP10, and TNFα levels and *AUD-related variables* (i.e., 1. AUD onset age, 2. AUD duration, 3. AUD DSM5 severity, 4. lifetime alcohol consumption, 5. days since last drink, 6. emergency-room detoxifications, 7. withdrawal-induced seizures, 8. AUD Identification Test (AUDIT) scores, 9. nicotine diagnosis, 10. history of smoking; requiring a Bonferroni-corrected *p*-value of 0.005 = 0.05/10 variables); *HIV-related variables* (i.e., 1. HIV onset age, 2. HIV duration, 3. CD4 cell count, 4. CD4 cell count nadir, 5. HIV viral load, 6. AIDS-defining events, 7. total number of HIV conditions, 8. ART medication status (on/off), 9. Karnofsky score, 10. VACS index; requiring a Bonferroni-corrected *p*-value of 0.005 = 0.05/10 variables); *HCV-related variables* (i.e., 1. HCV viral load, 2. FIB4 index, 3. APRI score, 4. AGR, 5. injection drug use, 6. AST, 7. ALT, 8. alkaline phosphatase (ALP), 9. γ-glutamyl transferase (GGT), 10. bilirubin; requiring a Bonferroni-corrected *p*-value of 0.005 = 0.05/10 variables); and *laboratory blood markers* (i.e., complete blood count: 1. hematocrit, 2. hemoglobin, 3. mean corpuscular volume, 4. red blood cells, 5. white blood cells; metabolic panel: 6. creatinine, 7. estimated glomerular filtration rate (eGFR), 8. glucose; nutrition panel: 9. folate, 10. prealbumin, requiring a Bonferroni-corrected *p*-value of 0.005 = 0.05/10 variables). Results of correlations are presented in Table 3. To summarize, there were no correlations between the levels of the 3 soluble proteins and AUD-related variables; ILβ levels were related to HIV-variables; IP10 levels were related to HCV-related variables and prealbumin; TNFα levels were related to both HIV and HCV related variables and several laboratory measures.

### 2.5. Principal Component Analysis

The 15 variables showing relationships with the 3 soluble proteins (from Table 3) were entered into a PCA (diagonals = 1, varimax rotation, limited to 3 factors) for data reduction for further analysis. Factor 1 (6 variables) included HCV viral load, FIB4, APRI, AST, ALT, and GGT. Factor 2 (5 variables) included injection drug use, VACS index, creatinine, eGFR, and prealbumin. Factor 3 (3 variables) included HIV viral load, total HIV conditions, and AGR. The principal components were saved with imputation for each subject.

### 2.6. Path Analysis

A path analysis including the 3 factors, the 3 soluble proteins, and subcortical white matter volume is presented in Figure 4. Factor 1 explained a large portion of variance (46%) in IP10 levels; factor 2 explained a similar proportion of the variance in TNFα levels. ILβ levels were also associated with factor 2. Among the soluble proteins, TNFα (20%) explained a larger portion of the variance in subcortical white matter volume than IP10 (8%) or ILβ (12%). In separate analysis for each soluble protein (Appendix A), similar relationships were observed: factor 2 best predicted ILβ and TNFα levels; factor 1 predicted IP10 levels. Again, TNFα (23%) relative to IP10 (15%) or ILβ (13%) explained a larger portion of the variance in subcortical white matter volume. Finally, relative to the 3 factors, TNFα explained the largest portion of the variance in subcortical white matter volume (Appendix A).

## 3. Discussion

Analysis of 3 patient (AUD, HIV, AUD+HIV) groups relative to healthy controls identified 3 soluble proteins as related to diagnoses: IL1β was low in the 3 patient groups regardless of HCV status; IP10 and TNFα levels were higher than controls in patients with HCV or HIV. The most novel finding herein is the correlation between higher peripheral TNFα levels and smaller subcortical white matter volume in patients with HCV or HIV. Relationships between peripheral proinflammatory cytokine levels and infarct size following cerebral ischemia have been reported in the literature for some time [57,58] with hints that TNFα may have a privileged status with respect to influencing brain integrity [59]. Indeed, a role for TNFα in subcortical white matter damage in HIV-associated dementia has been proposed since 1993 based on TNFα quantification in postmortem brain tissue [60,61,62,63,64]. A large study of 1926 healthy individuals found a relationship between higher circulating TNFα levels and smaller total brain volume [65,66,67]. Additionally, in healthy individuals (*n* = 303), carriers of a TNFα polymorphic variant had smaller hippocampal volumes than non-carriers [68,69,70]. In older depressed adults, higher TNFα levels are associated with a greater volume of white matter hyperintensities [71,72]. Further, in bipolar disorder, a panel of proinflammatory cytokines including TNFα was associated with loss of frontal white matter integrity [73]. Whether TNFα specifically affects brain white matter integrity in a causative manner requires longitudinal investigation in larger samples.

Additional correlates are consistent with the interpretation that changes in the levels of IL1β, IP10, and TNFα are essentially explained by HIV or HCV infection. For example, IP10 and TNFα correlated with the albumin/globulin ratio (i.e., AGR) and a low AGR can be indicative of infection, liver disease, or malnutrition [74,75,76,77,78]. Indeed, the majority of IL1β, IP10, and TNFα correlates described herein comport with a large body of literature indicating the mediating roles of proinflammatory cytokines in the pathogenesis of viral infections. Elevations in plasma IP10 [79,80,81,82] and TNFα [83,84,85] are frequently reported in the HIV literature and correlate with HIV viral load. The current study demonstrating a predilection for higher IP10 in HCV-related hepatic inflammation and fibrosis (i.e., correlations with HCV viral load, FIB4, APRI, and other measures of liver dysfunction such as elevated AST, ALT, and GGT) replicates previous experiments [86,87], confirms reports that IP10 may be a useful biomarker of HCV and liver integrity [88,89,90], and suggests that HIV+HCV may result in significantly higher levels of IP10 than in either infection alone [91]. To our knowledge, the relationship between higher IP10 and lower prealbumin levels has not previously reported in the literature, but prealbumin levels are low in liver disease [92,93,94,95,96].

Non-liver and non-infection-related correlates of TNFα identified in this study also feature in the extant literature. For example, HIV infection features a strong negative correlation between TNFα and hemoglobin levels [97]. As TNFα inhibits erythropoiesis, elevated TNFα may contribute to anemia in HIV [98,99,100,101]. TNFα is also elevated in kidney disease [102] and correlates with high creatinine and low eGFR [100,101,103]. 

Low IL1β associated with HIV viremia and HIV-related complications (e.g., bacteremia, leukoplakia, neuropathy) is an exception in the literature as IL1β is considered proinflammatory and levels are typically [104,105,106] but not always [107] elevated in HIV. IL1β levels are also typically elevated in liver disease [108,109,110].

The original hypothesis was based on an emerging body of literature and expected that proinflammatory cytokines would be elevated in AUD regardless of HCV status. The current results, however, suggest that levels of these cytokines correlate with active infection unrelated to AUD or HIV diagnosis per se; rather, levels of soluble proteins were related to variables associated with untreated HCV rather than treated HIV infection or uninfected AUD. A limitation of this study, therefore, is that it did not include a patient population with active HCV infection without comorbidity to test the hypothesis that elevated TNFα and IP10 levels would correlate with diminished brain integrity.

In conclusion, this study aimed to determine whether elevated levels of TNFα and IP10 in HIV or HCV infection were related to brain integrity measures. Contrary to our initial hypotheses but consistent with the available literature, only TNFα showed relations with brain selective to subcortical white matter volume.

## 4. Materials and Methods

### 4.1. Study Participants

This cross-sectional, observational study was conducted in accordance with protocols approved by the Institutional Review Boards of Stanford University and SRI International. Written informed consent was obtained from all participants in accordance with the Declaration of Helsinki by the signing of consent documents in the presence of staff after staff ensured that each participant understood the information provided and appreciated the reasonably foreseeable consequences of a participating in the study. Study participants were healthy controls (39 women/47 men, 54.5 ± 11.8 years), individuals with AUD (41 women/91 men, 53.0 ± 9.8 years; currently sober as demonstrated by a negative Breathalyzer test given immediately following consent), those infected with HIV (18 women/37 men, 57.7 ± 7.9 years), and those comorbid for HIV and AUD (18 women/30 men, 57.2 ± 6.8 years), for a total of 321 participants.

All participants were screened using the Structured Clinical Interview for Diagnostic Statistical Manual (SCID/DSM)-IV or DSM5 [111], structured health questionnaires, and a semi-structured timeline follow-back interview to quantify lifetime alcohol consumption [112]. AUD participants were recruited from local outpatient substance abuse treatment programs and met DSM-IV-TR criteria for alcohol dependence or DSM-5 criteria for AUD. HIV patients were referred from local outpatient or treatment centers or recruited during presentations by project staff and by distribution of flyers at community events. Comparison participants were recruited from the local community by referrals and flyers. Upon initial assessment, subjects were excluded if they had fewer than 8 years of education, or a significant history of medical (e.g., epilepsy, stroke, multiple sclerosis, uncontrolled diabetes, or loss of consciousness >30 min), psychiatric (i.e., schizophrenia or bipolar I disorder), or neurological (e.g., neurodegenerative disease) disorders other than alcohol abuse or dependence (DSM-IV) or AUDs (DSM5) in the alcoholic (i.e., AUD) group. Other exclusionary criteria were recent (i.e., past 3 months) substance dependence other than alcohol in the AUD group or any DSM-IV/DSM5 Axis I disorder in the control group. Severity of depressive symptoms was assessed with the Beck Depression Inventory-II (BDI-II) [113] in all groups.

### 4.2. Blood Sample Collection and Processing

Whole blood samples (~4 mL) were collected in lavender EDTA tubes between March 2013 and June 2019. Samples were centrifuged (500 relative centrifugal force (rcf) at room temperature for 10 min). Plasma was transferred to 1.5 mL conical tubes, centrifuged at 13,000 rcf at room temperature for another 10 min, and the resulting supernatant was transferred to 1.5 mL conical tubes for storage at −80 °C until analysis by the Stanford Human Immune Monitoring Center (HIMC). Additional blood samples (~40 mL) were collected and immediately analyzed by Quest Diagnostics (Secaucus, NJ, USA) for complete blood count with differential, comprehensive metabolic panel, HIV and HCV screening, and RNA quantification when relevant (i.e., for HIV or HCV seropositive individuals). Laboratory testing was not available for 16 control, 6 AUD, 5 HIV, and 4 AUD+HIV participants.

Laboratory results were used to calculate 2 non-invasive indices of liver injury: the Fibrosis index (FIB4: based on age, aspartate aminotransferase (AST), alanine aminotransferase (ALT), and platelet count) [114] and the AST/platelet count ratio (APRI) score, which have high predictive accuracy for diagnoses of liver fibrosis [115,116]. FIB4 and APRI formulas are available in our previous publication [33]. Further, the simple ratio of albumin to globulin (AGR) has been used to predict mortality risk related to HCV infection [117,118].

### 4.3. Immunological Assays

The HIMC (http://iti.stanford.edu/himc/ (accessed on 5 May 2021)) which continually benchmarks processes to minimize technical variability (Maecker et al., 2005), performed immunological assays. Human 41-plex kits (HCYTOMAG-60K, 11 kits) were purchased from EMD Millipore (Burlington, MA, USA) and used according to the manufacturer’s recommendations with modifications. Briefly, samples were mixed with antibody-linked magnetic beads on a 96-well plate and incubated overnight incubation at 4 °C with shaking. Cold and room temperature incubation steps were performed on an orbital shaker at 500–600 rpm. Plates were washed twice with wash buffer in a BioTek ELx405 washer (BioTek Instruments, Winooski, VT, USA). Following one hour incubation at room temperature with biotinylated detection antibody, streptavidin fluorochrome (i.e., streptavidin-PE) was added for 30 min with shaking. Plates were washed as described and phosphate-buffered saline (PBS) added to wells for reading in the Luminex 200 Instrument (Merck KGaA, Darmstadt, Germany) with a lower bound of 50–100 beads per sample per soluble protein. Each sample was measured in duplicate. Custom assay control beads by Radix Biosolutions (Georgetown, TX, USA) were added to all wells.

The 41 soluble proteins quantified belong to 4 families: ***hematopoietin*** (interleukin (IL)-IL1α, IL1β, IL-1RA, IL2, IL3, IL4, IL5, IL6, IL7, IL9, IL10, IL12-p40, IL12-p70, IL13, IL15, IL17, soluble CD40 ligand (CD40L), Fms-related tyrosine kinase 3 ligand (Flt3 ligand), granulocyte colony-stimulating factor (GCSF), granulocyte macrophage CSF (GMCSF)); ***chemokines*** (epidermal growth factor (EGF), eotaxin (CCL11), fibroblast growth factor (FGF)-2, fractalkine, RANTES (regulated on activation, normal T cell expressed and secreted/CCL5), growth regulated oncogene (GRO/CXCl1), IL8, Interferon-γ-induced protein 10 (IP10/CXCL10), monocyte chemoattractant protein 1 (MCP-1/CCL2), MCP-3 (CCL7), macrophage-derived chemokine (MDC/CCL22), macrophage inflammatory protein (MIP)-1α, MIP-1β, transforming growth factor (TGF)α, vascular endothelial growth factor (VEGF)); ***growth factors*** (platelet-derived growth factor (PDGF)AA, PDGFBB, Tumor Necrosis Factor α (TNFα), TNFβ); and ***interferons*** (IFNα2, IFNγ).

### 4.4. Brain Volumes

Brain imaging cases matched to date of plasma collection for each individual were extracted from a laboratory release of 1602 instances of analyzed structural MRI data. Brain volumes used in the current analyses were age and supratentorial volume (svol)-corrected based on a subset of 238 vetted healthy controls. The 238 controls used for age- and svol-correction comprised 121 men and 117 women; 158 white, 25 black, and 53 other races; aged 46.5 ± 17.1 years (age range 19.5 to 86.1 years); predominately right-handed (206 right-handed, 8 left-handed, and 9 ambidextrous); with an average BMI of 25.4 ± 4.1; average education of 16.3 ± 2.2 years; and an average SES of 24.3 ± 11.2.

### 4.5. Image Acquisition and Processing

MR data were collected and processed using an in-house pipeline as described [49]. Briefly, structural images of the brain were collected on a 3-Tesla GE whole-body MR system (General Electric (GE) Healthcare, Waukesha, WI, USA) using an 8-channel phased-array head coil. The axial T1-weighted Inversion-Recovery Prepared SPoiled Gradient Recalled (SPGR) sequences acquired in the 6-year interval (i.e., March 2013–June 2019) used similar (e.g., inversion time (TI) = 300 ms, matrix = 256 × 256, thickness = 1.25 mm, skip = 0 mm, 124 slices, field of view (FOV) = 24 cm) or negligibly different (e.g., repetition time (TR) = 6.55 or 5.92 ms, echo time (TE) = 1.56 or 1.93 ms) parameters.

Preprocessing of T1-weighted SPGR data involved noise removal [119] and brain mask segmentation using FSL BET [120], AFNI 3dSkullStrip [121], and Robust Brain Extraction (ROBEX) [122] generating 3 brain masks. In parallel, noise-corrected, T1-weighted images were corrected for field inhomogeneity via N4ITK [123] and brain masks were segmented using the 3 methods listed above plus FreeSurfer mri_gcut [124]. The resulting 7 segmented brain masks were reduced to one using majority voting [125].

Brain tissue segmentation (gray matter, white matter, and cerebrospinal fluid) of the skull-stripped T1-weighted images was generated via Atropos [123]. Parcellated maps of gray matter regions used the parc116 atlas to define cortical and subcortical regions summed for bilateral hemispheres. Five regions including the frontal cortex, subcortical white matter, caudate, putamen, and pallidum were evaluated to determine potential associations with soluble protein levels.

### 4.6. Statistical Analysis

Duplicate raw values of Mean Fluorescence Intensity (MFI) for each analyte were run through an R utility released by HIMC (http://iti.stanford.edu/himc/new-statistical-consultation-service.html (accessed on 12 February 2021)), which corrects for plate (i.e., batch/lot) and nonspecific binding artifacts [126]. Following corrections, the group mean MFI value for each soluble protein was added back to the detrended values. Additional statistics were performed using JMP Pro 14.1.0 (SAS Institute Inc., North Carolina, USA). Four-group diagnoses effects were evaluated using analysis of variance (ANOVA); two-group comparisons used *t*-tests. Diagnostic effects on soluble proteins were only considered significant for Bonferroni-corrected probability values of *p* ≤ 0.001 (i.e., 0.05/41 soluble proteins). Correlations were evaluated using nonparametric χ^2^ or Spearman’s *ρ*. Correlations with brain volumes were considered significant for Bonferroni-corrected probability values of *p* ≤ 0.01 (i.e., 0.05/5 brain regions). Other correlations required Bonferroni-corrected values as described in relevant Results section. A data-driven approach including variables associated with soluble proteins were entered into a JMP-based principal component analysis (PCA). The soluble proteins showing diagnoses effects, results of the PCA limited to 3 factors, and relevant brain regions were entered in a path analysis in R (version 3.2.4) [127].

## Figures and Tables

**Figure 1 ijms-22-04953-f001:**
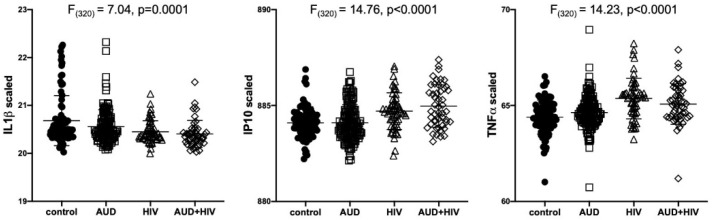
Levels of soluble proteins by initial diagnoses. Relative to the control group, IL1β levels were low in the AUD (*p* = 0.02), HIV (*p* = 0.0004), and AUD+HIV (*p* = 0.0001) groups. Relative to the control group, IP10 and TNFα levels were high in the HIV and AUD+HIV groups (both *p* < 0.0001). IL1β: Interleukin (IL)1β; IP10: Interferon γ-induced Protein 10 (IP10); TNFα: Tumor Necrosis Factor α; black circles: healthy controls; open squares: Alcohol Use Disorder (AUD); open triangles: Human Immunodeficiency Virus (HIV); open diamonds: AUD+HIV.

**Figure 2 ijms-22-04953-f002:**
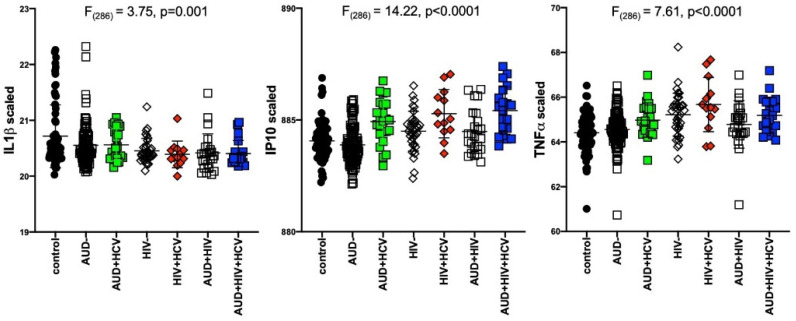
Levels of soluble proteins by hepatitis C virus (HCV) serostatus among patient groups. IL1β: relative to controls, lower in AUD (*p* = 0.008), HIV (*p* = 0.001), HIV+HCV (*p* = 0.007), AUD+HIV (*p* = 0.001), AUD+HIV+HCV (*p* = 0.002). IP10: relative to controls, higher in AUD+HCV (*p* < 0.0001), HIV (*p* = 0.02), HIV+HCV (*p* < 0.0001), AUD+HIV (*p* = 0.5), AUD+HIV+HCV (*p* < 0.0001). Higher in AUD+HCV vs. AUD (*p* < 0.0001), HIV+HCV vs. HIV (*p* = 0.009), AUD+HIV+HCV vs. AUD+HIV (*p* = 0.0008). black circles: healthy controls; open squares: AUD; filled squares: AUD+HCV; open triangles: HIV; filled triangles: HIV+HCV; open diamonds: AUD+HIV; filled diamonds: AUD+HIV+HCV.

**Figure 3 ijms-22-04953-f003:**
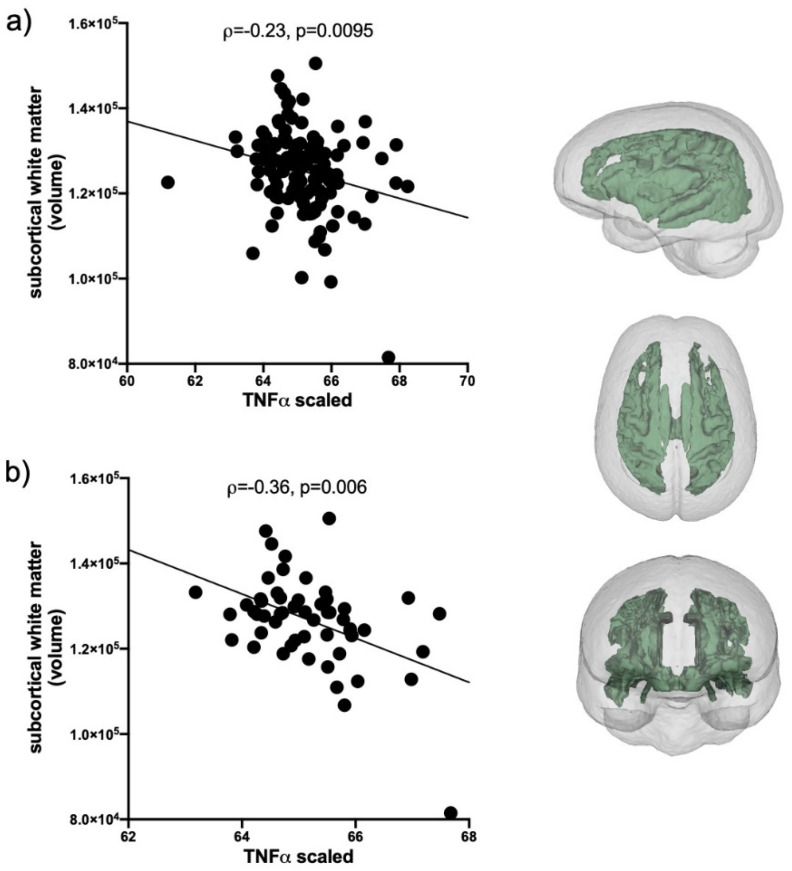
TNFα correlations with subcortical white matter volume. (**a**) In HCV seropositive AUD individuals and the 2 HIV groups (black circles, *n* = 127). (**b**) In the subset of HCV seropositive individuals collapsed across the 3 patient groups (black circles, *n* = 57). Inset: subcortical white matter volume.

**Figure 4 ijms-22-04953-f004:**
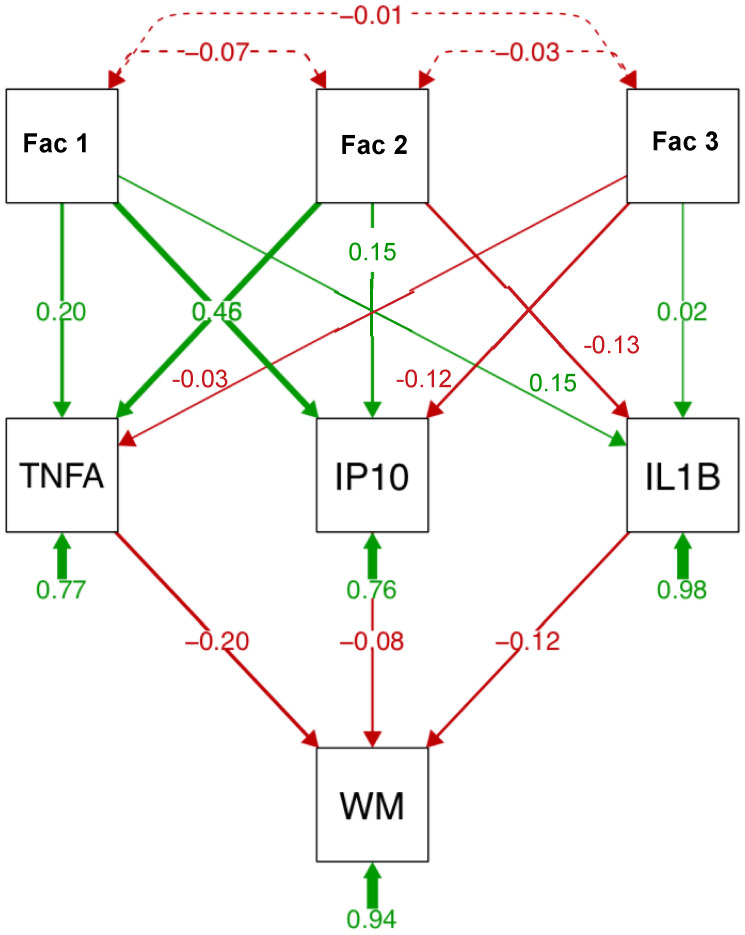
Path analysis describing relationships between relevant variables, soluble protein levels, and subcortical white matter volume (WM). Path analysis includes 3 factors (reduced from 15 variables entered into principal component analysis (PCA)), 3 soluble proteins, and subcortical white matter in the HCV and HIV groups (*n* = 127). Factor 1 (6 variables) includes HCV viral load, fibrosis score (FIB4), aspartate aminotransferase/platelet count ratio (APRI), aspartate aminotransferase (AST), alanine aminotransferase (ALT), and γ-glutamyl transferase (GGT). Factor 2 (5 variables) includes injection drug use (IDU), VACS index, creatinine, estimated glomerular filtration rate (eGFR), and prealbumin. Factor 3 (3 variables) includes HIV viral load, total HIV conditions, and the albumin-to-globulin ratio (AGR).

**Table 1 ijms-22-04953-t001:** Demographic characteristics of the 4 study groups: mean ± SD/frequency count.

	Control (*n* = 86)	AUD (*n* = 132)	HIV (*n* = 54)	AUD + HIV (*n* = 49)	*p*-Value *
**N (men/women)**	47/39	91/41	36/18	31/18	n.s.
**Age (years)**	54.5 ± 11.8	53.0 ± 9.8	57.7 ± 7.9	57.1 ± 6.6	**0.007**
**Self-Defined Ethnicity (Caucasian/AA ^a^/other ^b^)**	52/13/21	57/56/19	28/19/7	7/35/7	**<0.0001**
**Handedness (Right/Left/Ambidexterous)**	79/3/4	112/17/3	50/3/1	42/6/1	n.s.
**Body Mass Index**	26.1 ± 4.4	28.0 ± 4.8	26.2 ± 4.8	27.1 ± 4.5	**0.02**
**Education (years)**	16.4 ± 2.5	13.1 ± 2.5	13.9 ± 2.5	13.0 ± 2.3	**<0.0001**
**Socioeconomic Status ^c^**	23.5 ± 11.1	42.9 ± 15.8	36.8 ± 15.0	43.7 ± 13.2	**<0.0001**
**Global Assessment of Functioning**	85.6 ± 6.7	68.0 ± 10.2	71.7 ± 11.1	66.6 ± 8.9	**<0.0001**
**Smoker (never/past/current)**	80/2/4	30/31/71	30/11/13	15/10/24	**<0.0001**
**Beck Depression Inventory-II**	1.5 ± 2.3	8.8 ± 8.0	6.9 ± 6.1	8.7 ± 7.6	**<0.0001**
**WTAR IQ**	107.9 ± 10.8	96.9 ± 13.3	97.7 ± 13.6	89.9 ± 12.9	**<0.0001**
**AUD onset age**	-	25.4 ± 9.7	-	23.5 ± 8.7	n.s.
**Lifetime Alcohol Consumption**	-	1400.1 ± 1182.3	-	1140.3 ± 1001.0	n.s.
**Days since last drink**	-	234.2 ± 719.3	-	360.2 ± 1216.72	n.s.
**AUDIT ^d^ scores**	2.1 ± 1.7	17.8 ± 11.2	2.3 ± 2.5	10.2 ± 10.3	**<0.0001**
**HIV onset age (years)**		-	36.5 ± 10.1	35.0 ± 7.1	n.s.
**HIV duration (years)**	-	-	21.4 ± 8.1	22.2 ± 5.7	n.s.
**CD4 cell count (100/mm^3^)**	-	-	667.1 ± 256.6	660.6 ± 333.1	n.s.
**CD4 cell count nadir (100/mm^3^)**	-	-	196.3 ± 160.0	183.9 ± 176.4	n.s.
**Viral Load (log copies/mL)**	-	-	1.6 ± 0.8	1.9 ± 1.0	n.s.
**AIDS-defining event (yes/no) ^e^**	-	-	30/24	32/17	n.s.
**HAART (yes/no)**	-	-	49/5	45/4	n.s.
**Efavirinz, including Atripla (yes/no)**	-	-	2/52	4/45	n.s.
**VACS Index**	17.8 ± 13.3	18.5 ± 12.6	34.0 ± 17.4	32.0 ± 15.8	**<0.0001**
**Karnofsky score**	100.0 ± 0.0	99.8 ± 2.1	99.4 ± 3.1	98.8 ± 3.9	n.s.
**Hepatitis C Virus (positive/negative/missing)**	0/72/14	24/99/9	13/35/8	20/24/5	**<0.0001**
**Treatment for HCV infection ^f^ (yes/no/missing)**	-	7/120/5	4/48/2	5/43/1	n.s.

* 4-group comparisons: ANOVA used on continuous variables (e.g., age); χ^2^ used on nominal variables (e.g., handedness). ^a^ AA = African American; ^b^ other = Native American, Asian, Islander; ^c^ lower score = higher status; ^d^ AUDIT = Alcohol Use Disorders Identification Test; ^e^ including AIDS-defining illness or CD4 prior nadir < 200 cells/μL; ^f^ self report of HCV treatment; **bold** = significant.

**Table 2 ijms-22-04953-t002:** Soluble protein levels * in the 4 study groups: mean ± SD and ANOVA results.

Soluble Protein	Control (*n* = 86)	AUD (*n* = 132)	HIV (*n* = 54)	AUD + HIV (*n* = 49)	ANOVA	
					F Ratio	*p*-Value
**CD40L**	109.12 ± 0.41	109.19 ± 0.73	109.08 ± 0.75	108.97 ± 0.36	1.71	0.16
**EGF**	31.64 ± 0.51	31.52 ± 0.51	31.55 ± 0.86	31.33 ± 0.42	2.97	**0.03**
**EOTAXIN**	102.41 ± 0.55	102.42 ± 0.61	102.66 ± 0.64	102.48 ± 0.59	2.46	0.06
**FGFB**	22.07 ± 0.58	21.97 ± 0.46	21.91 ± 0.41	21.84 ± 0.5	2.43	0.07
**FLT3L**	34.14 ± 0.47	34.13 ± 0.41	34.28 ± 0.59	34.11 ± 0.45	1.67	0.17
**Fractaline**	19.61 ± 0.35	19.63 ± 0.32	19.6 ± 0.34	19.51 ± 0.28	1.49	0.22
**GCSF**	26.85 ± 0.41	26.74 ± 0.27	26.69 ± 0.31	26.75 ± 0.42	2.81	**0.04**
**GMCSF**	25.85 ± 0.28	25.81 ± 0.23	25.75 ± 0.20	25.74 ± 0.26	2.90	**0.04**
**GRO ^‡^**	935.26 ± 1.02	935.54 ± 0.97	935.37 ± 1.03	935.29 ± 1	1.66	0.18
**IFNA2**	20.8 ± 0.31	20.83 ± 0.41	20.77 ± 0.26	20.74 ± 0.26	0.94	0.42
**IFNG ^‡^**	51.86 ± 1.18	51.64 ± 0.88	51.65 ± 1.01	51.47 ± 0.94	1.72	0.16
**IL1A**	30.96 ± 0.44	30.81 ± 0.36	30.81 ± 0.31	30.76 ± 0.24	4.52	**0.004**
**IL1B**	20.68 ± 0.52	20.56 ± 0.35	20.45 ± 0.23	20.41 ± 0.28	7.04	**0.0001**
**IL1RA**	30.78 ± 0.4	30.76 ± 0.42	30.81 ± 0.50	30.72 ± 0.44	0.47	0.70
**IL2**	24.94 ± 0.51	24.85 ± 0.37	24.77 ± 0.27	24.74 ± 0.33	3.49	**0.02**
**IL3**	20.04 ± 0.37	20.01 ± 0.24	19.96 ± 0.18	19.93 ± 0.18	2.06	0.11
**IL4**	29.01 ± 0.42	28.95 ± 0.38	28.91 ± 0.35	28.77 ± 0.3	4.29	**0.01**
**IL5**	19.64 ± 0.51	19.54 ± 0.33	19.59 ± 0.55	19.5 ± 0.35	1.59	0.19
**IL6**	33.09 ± 0.62	33.01 ± 0.52	33.04 ± 0.58	32.86 ± 0.46	1.82	0.14
**IL7**	21.05 ± 0.38	21 ± 0.26	20.96 ± 0.2	20.96 ± 0.29	1.64	0.18
IL8	125.98 ± 0.61	125.94 ± 0.54	126.12 ± 0.62	126.07 ± 0.6	1.43	0.23
**IL9**	24.86 ± 0.52	24.76 ± 0.4	24.67 ± 0.37	24.62 ± 0.29	4.17	**0.007**
**IL10**	28.77 ± 0.49	28.72 ± 0.34	28.69 ± 0.36	28.66 ± 0.27	1.05	0.37
**IL12P40 ^‡^**	27.13 ± 1.25	26.88 ± 0.93	26.67 ± 0.61	26.81 ± 0.99	2.67	**0.05**
**IL12P70 ^‡^**	21.52 ± 1	21.51 ± 1.1	21.28 ± 0.71	21.32 ± 1	1.09	0.35
**IL13**	23.74 ± 0.62	23.6 ± 0.47	23.55 ± 0.51	23.47 ± 0.31	3.54	**0.02**
**IL15**	30.6 ± 0.38	30.55 ± 0.28	30.48 ± 0.18	30.5 ± 0.3	2.12	0.10
**IL17**	44.53 ± 0.73	44.35 ± 0.58	44.4 ± 0.64	44.15 ± 0.59	3.79	**0.01**
**IP10 ^‡^**	884.1 ± 0.83	884.1 ± 0.94	884.71 ± 0.96	884.97 ± 1.1	14.76	**<0.0001**
**MCP1 ^‡^**	966.42 ± 0.99	966.5 ± 1.01	966.71 ± 0.98	966.51 ± 1.01	0.93	0.43
**MCP3**	29.21 ± 0.8	29.01 ± 0.57	28.95 ± 0.61	28.95 ± 0.62	2.67	**0.05**
**MDC ^‡^**	692.47 ± 1	692.63 ± 0.95	692.54 ± 1.19	692.61 ± 0.92	0.49	0.69
**MIP1A**	97.21 ± 0.62	97.1 ± 0.59	97.19 ± 0.49	97.38 ± 1.1	2.08	0.10
**MIP1B**	58.64 ± 0.49	58.57 ± 0.44	58.6 ± 0.39	58.67 ± 0.67	0.68	0.56
**PDGFAA**	3394.5 ± 0.84	3394.66 ± 0.85	3394.63 ± 0.82	3394.55 ± 0.69	0.78	0.58
**PDGFBB**	636.33 ± 1.04	636.7 ± 0.89	636.51 ± 0.94	636.47 ± 0.89	2.78	**0.04**
**RANTES ^‡^**	7924.63 ± 1.08	7924.86 ± 1.12	7924.95 ± 0.73	7924.84 ± 0.72	1.46	0.23
**TGFA**	27.41 ± 0.46	27.4 ± 0.45	27.46 ± 0.63	27.39 ± 0.52	0.19	0.91
**TNFA ^‡^**	64.4 ± 0.89	64.63 ± 0.89	65.37 ± 1.06	65.08 ± 1.06	14.23	**<0.0001**
**TNFB**	27.78 ± 0.61	27.7 ± 0.52	27.66 ± 0.67	27.56 ± 0.45	1.69	0.17
**VEGF**	26.83 ± 0.52	26.75 ± 0.42	26.73 ± 0.6	26.64 ± 0.5	1.56	0.20

* R-utility corrected for plate and nonspecific binding artifacts; ^‡^ corrected for age, sex, race or combination (see text for details); **bold** = significant.

**Table 3 ijms-22-04953-t003:** Nonparametric Spearman’s *ρ* correlates of soluble protein levels.

Soluble Protein	AUD Related	HIV Related	HCV Related	Laboratory Meaures
IL1A		viral load: *ρ* = −0.23, *p* = 0.0007		
		HIV conditions: *ρ* = −0.17, *p* = 0.008		
IP10			IDU: χ^2^ = 17.6, *p* < 0.0001	
			viral load: *ρ* = 0.40, *p* = 0.0002	
			FIB4: *ρ* = 0.27, *p* = 0.004	
			APRI: *ρ* = 0.34, *p* = 0.0002	
			AGR: *ρ* = 0.34, *p* = 0.0002	
			AST: *ρ* = 0.43, *p* < 0.0001	
			ALT: *ρ* = 0.27, *p* = 0.003	
			GGT: *ρ* = 0.26, *p* = 0.005	
				prealbumin: *ρ* = −0.31, *p* = 0.0007
TNFA		VACS index: *ρ* = −0.35, *p* = 0.0004		
			FIB4: *ρ* = 0.27, *p* = 0.004	
			AGR: *ρ* = −0.36, *p* < 0.0001	
				hemoglobin: *ρ* = −0.26, *p* = 0.005
				creatinine: *ρ* = 0.27, *p* = 0.004
				eGFR: *ρ* = −0.32, *p* = 0.003

Abbrevations: injection drug use (IDU), fibrosis score (FIB4), aspartate aminotransferase/platelet count ratio (APRI), albumin/globulin ratio (AGR), aspartate aminotransferase (AST), alanine aminotransferase (ALT), *γ*-glutamyl transferase (GGT), estimated glomerular filtration rate (eGFR).

## Data Availability

The data that support the findings of this study will be openly available (12 October 2021) at https://data.mendeley.com/ in Zahr cytokine brain.

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
