# Peer review of "Preliminary Evidence for a Relationship between Elevated Plasma TNFα and Smaller Subcortical White Matter Volume in HCV Infection Irrespective of HIV or AUD Comorbidity"

_ijms, 2021, doi:10.3390/ijms22094953_

Round 1

Reviewer 1 Report

File attached.

Author Response

Abstract:

Page 1, line 16: "Soluble protein levels were quantified in 86 healthy controls, 132 individuals with AUD, 54 individuals seropositive for human immunodeficiency virus (HIV), and 49 individuals with AUD and HIV and correlated to regional brain volumes as quantified with structural magnetic resonance imaging (MRI)." This statement is confusing and gives no information on the HCV positive patients.

The abstract has been modified as follows:

“Among the patient groups, HCV was present in 24 of the individuals with AUD, 13 individuals with HIV, and 20 of the comorbid AUD and HIV group.”

Page 2, line 50: Introduction: "Our previous work reported higher than control IP10 and TNFα plasma levels in abstinent AUD individuals seropositive for hepatitis C virus (HCV) infection and in individuals infected with the human immunodeficiency virus (HIV)." In the abstract HIV patients are not mentioned in this context.

The abstract has been modified as follows:

“Classical inflammation in response to bacterial, parasitic, or viral infections such as HIV includes local recruitment of neutrophils and macrophages and the production of proinflammatory cytokines and chemokines. Proposed biomarkers of organ integrity in Alcohol Use Disorders (AUD) include elevations in peripheral (plasma) levels of proinflammatory proteins. In testing this proposal, previous work included a group of HIV infected individuals as positive controls and identified elevations in the soluble proteins TNF and IP10; these cytokines were only elevated in AUD individuals seropositive for hepatitis C infection (HCV).”

Figure 1, 2. Please mark the statistical significance on each graph.

Statistical significance has been added to each graph.

Given that the individual differences of the measured 3 protein levels are very small (Table 2), would it be possible to predict anything (e.g. a smaller subcortical white matter volume) based on a certain value of any of these factors measured in a patient? One would expect high variability of TNF alpha in different patients and in different diseases.

Our original hypotheses based on an emerging literature expected that cytokines would be elevated in the AUD group regardless of HCV status. In addition to the finding that proinflammatory cytokines IP10 and TNFa were only elevated in alcoholics infected with HCV, however, the current results suggest that levels of these cytokines correlate with active infection. That is, variables corresponding with elevated IP10 and TNFa were related more to liver integrity (i.e., untreated HCV) than to ART-treated HIV infection.

This study, therefore, did not include a suitable patient population to test the hypothesis that elevated TNFa would correlate with smaller brain white matter volume. Our new hypothesis is that only patients with active infection or liver disease will have elevated TNFa levels. Future studies focused on patient populations with more reliably elevated TNFa levels will better determine whether this cytokine is a suitable marker for brain integrity.

Reviewer 2 Report

Thanks for the opportunity to review the manuscript #ijms-1202208 titled "Preliminary Evidence for a Relationship between Elevated Plasma TNFα and a Subcortical White Matter Volume Deficit in HIV and HCV Infection" proposed by Zahr and colleagues. The manuscript is well written.

I have a few comments that can be found below.

Title:

  1. An "s" should be added to "infection". The authors should highly consider to include AUD in the title.

Abstract:

  1. The background should include HIV. It is also not clear to the reader what the authors are interested in these associations.
  2. Information about study design is missing.

  1. There are too many keywords, including some that are not really "keywords".

 Introduction

  1. Line 32: "damage [2] or nutri-". "and" should be used instead of "or"
  2. Line 34 and throughout the manuscript: cite the references properly, e.g., delete cf. from [cf., 4].
  3. Lines 36-42: too long sentence. Consider splitting it.
  4. Lines 50-52 need a reference. The authors should avoid stacking references.
  5. Lines 57-59: rephrase this as a study hypothesis, and moved it towards the end of the section, before Lines 76. Change Lines 76-79 into study aim.

Materials and Methods

  1. Information about study design is missing.
  2. Informed consent seems too detailed.
  3. Eligibility criteria are not clearly stated.
  4. Line 118: (~4cc): change this into international units, e.g. mL.
  5. Throughout the manuscript: leave a space between a value and its unit.
  6. Lines 136-149: a good portion of the text looks different in color.
  7. Manufacturers and countries of manufacture should be provided throughout the section.

Results

  1. Is "4. Group Differences in Soluble Protein Levels" not results? Why is it a separate section?
  2. No need to highlight p values with significance.

  1. Figure titles should be improved, and the authors should separate legends and titles.

Discussion

  1. Any study limitations?

Author Response

Title:

  1. An "s" should be added to "infection". The authors should highly consider to include AUD in the title.

Title modified as suggested.

“Preliminary Evidence for a Relationship between Elevated Plasma TNF and a Subcortical White Matter Volume Deficit in HIV Infection and HCV Infection in AUD”

Abstract:

  1. The background should include HIV. It is also not clear to the reader what the authors are interested in these associations.

The abstract has been modified as follows:

“Classical inflammation in response to bacterial, parasitic, or viral infections such as HIV includes local recruitment of neutrophils and macrophages and the production of proinflammatory cytokines and chemokines. Proposed biomarkers of organ integrity in Alcohol Use Disorders (AUD) include elevations in peripheral (plasma) levels of proinflammatory proteins. In testing this proposal, previous work included a group of HIV infected individuals as positive controls and identified elevations in the soluble proteins TNF and IP10; these cytokines were only elevated in AUD individuals seropositive for hepatitis C infection (HCV).”

  1. Information about study design is missing.

The abstract has been modified as follows:

The current, observational, cross-sectional study evaluated whether higher levels of these proinflammatory cytokines would be associated with compromised brain integrity.

  1. There are too many keywords, including some that are not really "keywords".

 The number of keywords has been reduced.

 Introduction

  1. Line 32: "damage [2] or nutri-". "and" should be used instead of "or"

Modified as suggested.

  1. Line 34 and throughout the manuscript: cite the references properly, e.g., delete cf. from [cf., 4].

Removed as suggested.

  1. Lines 36-42: too long sentence. Consider splitting it.

Split as suggested.

  1. Lines 50-52 need a reference. The authors should avoid stacking references.

Reference added and stacking references modified.

  1. Lines 57-59: rephrase this as a study hypothesis, and moved it towards the end of the section, before Lines 76. Change Lines 76-79 into study aim.

The hypotheses and aims are clear in their current format.

 Materials and Methods

  1. Information about study design is missing.

Added the following to the Participants section:

“This cross-sectional, observational study was conducted in accordance with protocols approved by the Institutional Review Boards of Stanford University and SRI International.”

  1. Informed consent seems too detailed.

Prefer to include consent details.

  1. Eligibility criteria are not clearly stated.

Added that AUD participants “met DSM-IV-TR criteria for alcohol dependence or DSM-5 criteria for AUD.” Additional eligibility criteria are detailed in Table 1.

  1. Line 118: (~4cc): change this into international units, e.g. mL.

Changed units from cc to mL.

  1. Throughout the manuscript: leave a space between a value and its unit.

Done.

  1. Lines 136-149: a good portion of the text looks different in color.

Modified.

  1. Manufacturers and countries of manufacture should be provided throughout the section.

Added.

Results

  1. Is "4. Group Differences in Soluble Protein Levels" not results? Why is it a separate section?

This heading was introduced by editors and has been modified back to author’s original formatting.

  1. No need to highlight p values with significance.

Uploaded new tables with no highlighting.

  1. Figure titles should be improved, and the authors should separate legends and titles.

Modified.

Discussion

  1. Any study limitations?

Added the following limitation:

“The original hypothesis was based on an emerging literature and expected that proinflammatory cytokines would be elevated in AUD regardless of HCV status. The current results, however, suggest that levels of these cytokines correlate with active infection unrelated to AUD or HIV diagnosis per se; rather, levels of soluble proteins were related to variables associated with untreated HCV rather than treated HIV infection or uninfected AUD. A limitation of this study, therefore, is that it did not include a patient population with active HCV infection without comorbidity to test the hypothesis that elevated TNF and IP10 levels would correlate with diminished brain integrity.”